# Evaluation of Volatile Metabolites Emitted In-Vivo from Cold-Hardy Grapes during Ripening Using SPME and GC-MS: A Proof-of-Concept

**DOI:** 10.3390/molecules24030536

**Published:** 2019-02-01

**Authors:** Somchai Rice, Devin L. Maurer, Anne Fennell, Murlidhar Dharmadhikari, Jacek A. Koziel

**Affiliations:** 1Midwest Grape and Wine Industry Institute, Iowa State University, Ames, IA 50011, USA; somchai@iastate.edu (S.R.); murli@iastate.edu (M.D.); 2Interdepartmental Toxicology Graduate Program, Iowa State University, Ames, IA 50011, USA; 3Department of Agricultural and Biosystems Engineering, Iowa State University, Ames, IA 50011, USA; dmaurer@iastate.edu; 4Department of Agronomy, Horticulture and Plant Science, BioSNTR, South Dakota State University, Brookings, SD 57006, USA; anne.fennell@sdstate.edu

**Keywords:** biogenic emissions, veraison, viticulture, nondestructive analysis, wine aroma, diffusion, grape skin, vacuum-assisted extraction, solid-phase microextraction, VOCs

## Abstract

In this research, we propose a novel concept for a non-destructive evaluation of volatiles emitted from ripening grapes using solid-phase microextraction (SPME). This concept is novel to both the traditional vinifera grapes and the cold-hardy cultivars. Our sample models are cold-hardy varieties in the upper Midwest for which many of the basic multiyear grape flavor and wine style data is needed. Non-destructive sampling included a use of polyvinyl fluoride (PVF) chambers temporarily enclosing and concentrating volatiles emitted by a whole cluster of grapes on a vine and a modified 2 mL glass vial for a vacuum-assisted sampling of volatiles from a single grape berry. We used SPME for either sampling in the field or headspace of crushed grapes in the lab and followed with analyses on gas chromatography-mass spectrometry (GC-MS). We have shown that it is feasible to detect volatile organic compounds (VOCs) emitted in-vivo from single grape berries (39 compounds) and whole clusters (44 compounds). Over 110 VOCs were released to headspace from crushed berries. Spatial (vineyard location) and temporal variations in VOC profiles were observed for all four cultivars. However, these changes were not consistent by growing season, by location, within cultivars, or by ripening stage when analyzed by multivariate analyses such as principal component analysis (PCA) and hierarchical cluster analyses (HCA). Research into aroma compounds present in cold-hardy cultivars is essential to the continued growth of the wine industry in cold climates and diversification of agriculture in the upper Midwestern area of the U.S.

## 1. Introduction

Understanding the development of flavor and aroma compounds in wine grapes is crucial to winemaking. Grape berry development is characterized by two sigmoidal growth periods. The first growth period is berry formation from fruit set to lag phase. This is followed by berry-ripening from veraison to harvest [1]. Veraison is characterized by a change in color of the berries. During the berry ripening phase, sugar accumulates as measured in Brix. The rapid accumulation of sugar in the berry ripening from veraison onto harvest is well understood [2]. This is contrasted by the relative lack of research on aroma compound accumulation during ripening, especially for cold-hardy grapes. Further understanding of the accumulation of aroma compounds during the ripening phase can inform how viticultural practices can be used to influence wine style. 

Interest in research of aroma compounds in wine grapes is high. The prevailing share of published research in wine grapes has been in *V. vinifera* (‘Old World’, well-established varieties). This is expected since vinifera was cultivated as early as the seventh and fourth millennia BC [3]. For example, it well-known that aroma compounds such as pyrazines contribute to the characteristic aroma in Cabernet Sauvignon and Sauvignon Blanc [4,5]. These aromas can be described as ‘grassy,’ ‘herbaceous’, and ‘green bell pepper’ [6]. The decline in pyrazines in developing wine grapes has been linked to the levels of sunlight reaching the cluster and can be reduced through canopy management if this aroma is undesirable [7]. Terroir has been shown to affect wine aroma in Riesling grown in the Niagara Peninsula [8] and Cabernet Sauvignon in China [4]. Aroma compounds have been characterized in Japanese Shine muscat (*V. labruscana* Bailey and *V. vinifera* L.). Levels of linalool, hexanal, (*E*)-2-hexenal, hexanol, and (*Z*)-3-hexene-1-ol, nerol in berry skins and pulp were influenced by storage temperatures post-harvest [9]. Viticultural practices such as the timing of early defoliation have been investigated to determine the effect on Tempranillo wine aroma [10], and enological practices such as effects of pre-fermentation cold soaking on Cabernet Sauvignon grape and wine volatiles [11]. The bulk of the research into the aroma of grapes and wine has been done for vinifera because vinifera has existed longer than hybrid grapes. With the recent introduction of cold-hardy (hybrid) grapes, production of quality wines is possible in cold-climate regions where vinifera cannot thrive.

Since the release of high-quality, cold-hardy, and disease-resistant cultivars from the University of Minnesota, the winemaking industry has grown in cold climates such as the upper Midwest region of the U.S. St. Croix, Frontenac, Marquette, and La Crescent cultivars were developed in 1983, 1996, 2006, and 2002, respectively [12]. Current searches in the journal database Web of Science using keywords and variations of ‘Marquette,’ ‘Frontenac,’ ‘St. Croix’, ‘La Crescent’, volatile, aroma, cold-hardy, and maturity yield < 50 articles. Canopy management effects on fruit and wine aroma have been investigated in Traminette, an interspecific hybrid of Gewürztraminer in the Eastern U.S. [13]. Volatile compounds from Zuoshanyi, a native red grape variety in northeast China, were characterized with 135 VOCs identified and quantified [14]. Effects of pre-fermentation treatments on wine aroma profile were explored in the cold-hardy cultivar Solaris in Denmark [15]. 

There is a gap in knowledge, especially in aroma research, with these interspecific, cold-hardy hybrid grapes. Previous work showed a constant decrease in the ratio of alpha-linolenic acid degradation products, *cis*-3-hexenol to *trans*-2-hexenal during ripening of Frontenac and Marquette berries grown in Quebec, through the destructive blending of the berries [16]. Frontenac and Marquette aromas reported at harvest were mainly hexanal, *trans*-2-hexenal, 1-hexanol, *cis*-3-hexenol, hexanoic acid, acetic acid, *beta*-damascenone, and 1-phenylethanol. Marquette had significantly higher levels of linalool, geraniol, and *alpha*-citral [17]. Continuing work in Canada has been done in profiling aroma compounds in Frontenac, Marquette, Marechal Foch, Sabrevois, and St. Croix skin, juice and wine. Terpenes were primarily located in the skin, and the highest concentration was in Marquette. Nonanal, (*E*,*Z*)-2,6 nonadienal, *beta*-damascenone, ethyl octanoate, and isoamyl acetate were compounds with the highest odor activity values (OAV) in wines [17]. The OAV for a compound is the ratio between the concentration and the odor detection threshold (ODT) and it could be a useful metric for aroma-imparting compounds. The ODT is the minimal concentration that can be detected by human nose in 50% of the population [18,19,20]. Earlier research has also shown that the majority of aroma compounds present in grape berries are bound to a sugar moiety within the berry [21].

Various methods of sample preparation have been used to characterize aromas from grapes and wine. Thermal desorption was used to determine volatiles from Solaris wine [15]. Solid phase extraction (SPE) has been used to isolate aroma precursors in Merlot, Gewürztraminer, and Tempranillo grapes and wine [22]. Solid-phase microextraction (SPME) has emerged as one of the preferred methods of sample and sample prep for analysis of volatiles in the grapes and wine. SPME offers the advantages of portability, simplicity, and re-usability in field and laboratory settings. Applications using SPME in the food and beverage industry can be found elsewhere [23]. Gas chromatography (GC) has been extensively used to separate aroma compounds from the complex mix of aromas. GC is often coupled with mass spectrometry (MS) to identify and quantify the separated aroma compounds. These analytical methods have been used, sometimes in combination with other analytical methods, in analysis of volatiles from grapefruit (*Citrus paradise* L.) [24], berry cactus (*Myrtillocactus geometrizans*) [25], Shine Muscat [9], Cabernet Sauvignon [4,26], Zuoshanyi grapes [14], Muscat cultivars [27], Nero d’Avola and Fiano grapes [28], Monastrell wines [29], and selected cold-hardy grape cultivars and wine [16,17] and cold-hardy wines [17,30,31,32,33,34]. 

A review of SPME use for in-vivo and in-vitro in whole plant and plant organ analysis is found elsewhere [35]. It is clear that there is little research in cold-hardy wine grape cultivars when compared to *V. vinifera*. There is a need for a better understanding of these new cultivars in order to produce quality cold-climate wines that can compete in the world market. To date, this research is the first report of aroma compounds (1) emitted in-vivo from veraison to harvest using two novel sampling methods and collected by SPME and analyzed by GC-MS from Frontenac, Marquette, St. Croix, and La Crescent grape cultivars.

In this research, we propose a novel concept for a non-destructive evaluation of volatiles emitted from ripening grapes using SPME. This concept is novel to both the traditional vinifera grapes and the cold-hardy cultivars. Our models are cold-hardy varieties in the upper Midwest for which many of the basic multiyear grape flavor and wine style data is needed. Research into aroma compounds present in cold-hardy cultivars is essential to the continued growth of the wine industry in cold climates and diversification of agriculture in the upper Midwestern area of the U.S. The need for data is confounded by the small resources available to conduct long-term research. 

If proven feasible, the concept of non-destructive analysis of ripening grapes presents a tantalizing possibility to investigate the effects of different viticulture practices throughout the stages of berry ripening on berry aroma. This, in turn, could be used to develop better quality wines. If volatile compounds emitted in-vivo could be identified as developmental biomarkers, portable target VOC detectors could then be developed. These detectors can give vineyards a real-time gauge to guide them in harvesting for flavor.

The main objective was to develop the proof-of-concept for a non-destructive (in-vivo) sampling of volatile compounds from growing and ripening grapes. Specific objectives (1–4) were to (1) develop sampling devices to capture volatiles emitted from a whole cluster and single berry; (2) characterize the volatile compounds emitted in-vivo from four cold-hardy grape cultivars using: (2i) whole cluster analysis, (2ii) single berry analysis; (3) compare volatile compounds emitted in-vivo (objective 1) with crushed berry (i.e., destructive analysis including skin, seeds, and pulp); (4) search for preliminary links between volatile compounds detected (objectives 1 and 2) and selected: (4i) microclimates (Iowa and South Dakota), (4ii) the individual cultivars (i.e., Frontenac, Marquette, St. Croix, and La Crescent), and (4iii) time stages of berry ripening. 

The working hypotheses were: (1) aroma compound development from veraison-to-harvest can be detected in-vivo by sampling volatile emissions from ripening grapes (from both a single berry and whole grape cluster) and (2) that the flavor accumulation (i.e., increasing concentration of VOCs) can be correlated with berry ripening in all four cultivars. Testing these hypotheses can potentially translate into improving viticulture practices that lead to timing the harvesting for flavor. This research aims at addressing the gap in knowledge for cold-hardy grape cultivars by cataloging VOCs from Frontenac, Marquette, St. Croix and La Crescent emitted in-vivo and whole crushed berries throughout berry ripening.

## 2. Results

### 2.1. Sampling Devices to Capture Volatiles Emitted from a Whole Cluster and Single Berry

In this research, non-destructive and destructive sampling methods for the detection of VOCs emitted from cold-hardy grapes were explored. Non-destructive sampling included (1) a use of polyvinyl fluoride (PVF) chambers temporarily enclosing and concentrating volatiles emitted by a whole cluster of grapes on a vine (Figure 1), and (2) a modified 2 mL glass vial for a vacuum-assisted sampling of volatiles from a single berry (Figure 2). 

A total of 124 VOCs were identified across all sampling methods, 79 of these VOCs were verified with analytical standards matching retention times and mass spectral data (i.e., using the identification of compounds with Automated Mass Spectral Deconvolution and Identification System (AMDIS) target library search with at least 80% mass spectral match. Target libraries included (a) the 6 libraries that are included with the AMDIS program, (b) an onsite (our laboratory) library created from analysis of pure standards (200+ compounds), (c) NIST11 mass spectral library described in Materials and Methods section on data analysis). A full summary of VOCs identified in Frontenac, Marquette, St. Croix, and La Crescent berries from South Dakota and Iowa by each sampling method is provided in data paper [36] with known aroma descriptors for pure compounds [37,38]. All PCA biplots are given in Appendix B (Figure A1 and Figure A2). It should be noted that significant changes in volatiles emitted were only observed in Frontenac grapes grown in South Dakota in 2013 as indicated by the variance accounted for in component 1 and 2 in PCA (i.e., greater than 70%). However, due to the exploratory nature of this research, all PCA data is presented is subsequent sections. 

### 2.2. Volatiles Emitted In-Vivo from Four Cold-Hardy Grape Cultivars

PVF chambers were used in 2012 on Frontenac and Marquette in Iowa and South Dakota. Modified glass vials were used in 2013 on Frontenac, Marquette, St. Croix, and La Crescent in South Dakota. Only St. Croix and La Crescent were sampled by modified glass vials in Iowa, limited by funding. 

#### 2.2.1. Emissions from Whole Grape Cluster

Forty-four of the total 124 grape VOCs emitted in-vivo were detected by whole grape cluster sampling chambers in Frontenac and Marquette cultivars grown in Iowa and South Dakota, monitored from veraison to harvest. Table 1 presents the VOCs that are characteristic of biogenic emissions from Frontenac and Marquette clusters during the 2012 growing season from Iowa and South Dakota. These volatiles were detected in-vivo from whole grape clusters and determined through interpretation of principal component analysis (PCA) and hierarchical clustering analysis (HCA). A detailed summary of all 124 VOC can be found elsewhere [36]. However, only key representative volatiles from the hierarchical cluster analysis (HCA) are labeled with numbers on PCA biplot figures presented in Results.

##### Frontenac

3-Methyl-1-butanol and heptanal were emitted and detected in 2012 Iowa Frontenac grapes at veraison. At harvest, 1,4-butanolide was detected (Figure A1). Nonanal, benzyl alcohol, and toluene were emitted and detected in 2012 South Dakota Frontenac grapes (Figure 3) at veraison. Only one compound, i.e., acetic acid, was associated with harvest time. Other compounds were detected (e.g., 2-methyl-3-penten-2-one and 3-phenyl-2-propenal) but were not indicated to be the most representative compounds from HCA. Results are also presented in this manner throughout the manuscript.

##### Marquette

2012 Iowa Marquette did not have a ‘representative’ VOC at veraison, as indicated by HCA, and replicate samples had high variability (i.e., unevenly distributed between 2 quadrants of the PCA biplot). By harvest, 1-hexadecanol and methyl ethyl ketone had developed. Similarly, 2012 South Dakota Marquette VOCs emitted at veraison (Figure 4) did not have ‘representative’ VOC at veraison. At harvest, indene was the representative VOC emitted. 

#### 2.2.2. Emissions from Single Berries

Thirty-nine VOCs emitted in-vivo were also detected by modified glass vial (vacuum assisted) method in Frontenac, Marquette, St. Croix, and La Crescent cultivars grown in Iowa and South Dakota. Table 2 presents the VOCs that are characteristic of these 4 cold-hardy cultivars during the 2013 growing season in Iowa and South Dakota, detected in-vivo from single berries, and determined through multivariate statistical analysis previously discussed.

##### Frontenac

In-vivo detection of VOCs by modified glass vial did not identify a key representative compound in 2013 South Dakota Frontenac grapes at veraison (Figure 5). At harvest, palmitic acid was emitted and detected in these berries.

##### Marquette

VOCs detected by modified glass vial emitted from 2013 Marquette grown in South Dakota generally did not vary during berry development. The variability decreased between the replicate samples, indicated by less spread between the data points as the berries developed. Aromas from berries grown in South Dakota during the 2013 growing season (Figure 6) can be characterized from 3 VOCs. These compounds were 2-ethyl-1-hexanol, ethyl octanoate and 1,4-butanolide.

##### St. Croix

Statistical analysis of VOCs detected from 2013 St. Croix grown in Iowa at veraison and harvest determined 3 important compounds. These compounds were 3-methyl indole, benzyl alcohol, and acetic acid. Decreased variability between replicate samples was observed as the berries ripened, although no strong associations were noticed between these compounds and berry development. Compounds emitted and detected in 2013 St. Croix from South Dakota (Figure 7) were 1,4-butanolide, 5-(hydroxymethyl)-2-furancarboxaldehyde, ethyl acetate, and nonanal. Of the 5 key VOCs detected in 2013 South Dakota St. Croix at veraison, nonanal was most associated with development at harvest.

##### La Crescent

VOCs detected by modified glass vial emitted from 2013 La Crescent grown in Iowa were highly variable at veraison. A characteristic compound (i.e., 3-methyl-indole) was determined to be present at veraison. By harvest, octanal was present but not statistically representative. La Crescent berries from 2013 grown in South Dakota (Figure 8) were highly variable between replicate samples. Compounds emitted included 2-phenylethanol, 6-methyl-5-hepten-2-one, and acetic acid. By harvest, 1,4-butanolide was present but not statistically representative.

### 2.3. Destructive Sampling

117 grape VOCs were detected by destructive analysis (i.e., crushed berries) in Frontenac, Marquette, St. Croix, and La Crescent cultivars grown in Iowa and South Dakota. The sample matrix included skins, pulp, and seeds. Crushed berry analysis was used in 2012 on Frontenac and Marquette cultivars grown in South Dakota, and all 4 cultivars in 2013. A freezer malfunction in resulted in the loss of Iowa 2012 berries stored for crushed berry analysis. Table 3 presents the VOCs that are characteristic of these 4 cold-hardy cultivars during the 2012 and 2013 growing seasons in Iowa and South Dakota, detected in whole, crushed berries and determined through multivariate statistical analysis previously discussed.

#### 2.3.1. Frontenac

VOCs detected after crushing the berries of Frontenac grapes from the 2013 growing season in Iowa were isovaleraldehyde and isoamyl acetate at veraison. At harvest, VOCs were cyclohexanol and) and 3-methyl-1-butanol, as shown in Figure A1. VOCs detected after crushing berries of Frontenac grapes from the 2012 growing season in South Dakota were acetaldehyde and 1-hexanol at veraison in 2012. Frontenac grapes from the 2013 growing season in South Dakota was associated with ethyl hexanoate. VOCs detected after crushing berries of Frontenac grapes from the 2012 growing season at harvest in South Dakota were associated with alkane and styrene. In 2013 at harvest, however, compounds emitted were acetone, ethyl palmitate, hexanoic acid, 2-methyl-1-propanol, and 2-octanone in Frontenac grapes in South Dakota, Figure 9.

#### 2.3.2. Marquette

Compounds emitted from Marquette grapes from the 2013 growing season in Iowa were formic acid, octyl ester at veraison. By harvest, 2013 Marquette grapes emitted benzophenone, hexanal, and isoamyl acetate, Figure A1. In the 2012 South Dakota growing season, compounds such as cyclohexanol and (*E*)-2-hexenoic acid were most associated with Marquette berries at veraison. By harvest, these compounds shifted to styrene, beta-cyclocitral, and nonanal (Figure 10). 

#### 2.3.3. St. Croix

VOCs from crushed St. Croix grapes from the 2013 Iowa growing season changed from benzene acetaldehyde, isobutyraldehyde, ethyl butyrate, 1-Hexanol, beta-Damascenone, valeraldehyde, ethyl decanoate, methacrolein, 1-butanol, aspirin methyl ester at veraison to formic acid, and octyl ester and isoamyl acetate at harvest (Figure A1). VOCs from crushed St. Croix grapes from the 2013 growing season in South Dakota changed from benzyl alcohol, benzaldehyde, and N-benzyl-2-phenethylamine at veraison to ethyl acetate, methyl salicylate, safrol, propionaldehyde, 2-phenylethanol, 1-Pentanol, 2-heptanone, benzoic acid, and methyl ester at harvest (Figure 11).

#### 2.3.4. La Crescent

VOCs from La Crescent berries from the 2013 Iowa growing season changed from propanoic acid, ethyl butyrate, 3-methyl-1-butanol, beta-cyclocitral at veraison to p-cymene, beta-damascenone, 1-hexanol, and beta-pinene at harvest (Figure A1). In the 2013 South Dakota growing season, La Crescent VOCs from crushed berries changed from isoamyl acetate, linalyl acetate, 2-ethyl-1-hexanol, geraniol, isophorone, and allyl alcohol at veraison to ethyl butyrate, propyl-benzene, and styrene at harvest (Figure 12).

## 3. Discussion

The effect of vineyard practices on grape and wine aroma merit study. The claims to a regions’ wines by sensory attributes need to be scientifically correlated to bolster the local economies. Otherwise consumers are inundated with marketing claims. This study attempted to compare microclimates of Iowa and South Dakota during 3 months of the growing season, over 2 years. The Iowa plot is in USDA plant hardiness zone 5a [39]. In comparison, the South Dakota plot is in USDA plant hardiness zone 4b [39]. These metrics were obtained from the National Oceanic and Atmospheric Administration, formerly the National Climatic Data Center [40]. 

Preliminary analyses using ANOVA were also completed (shown in ‘ANOVA in-vivo’ spreadsheet, Appendix A). Type 1 sum of squares analysis indicated statistical significance of method, cultivar, date and time, method and cultivar interaction, and method and site and cultivar interactions (shown in Table A1). Post-hoc Tukey HSD test is (Table A2) shows differences within a method, cultivar, date and time, method and cultivar interaction, and method and site and cultivar interactions. PCA analyses were used to determine key volatile compounds emitted. Although significant differences were noted using ANOVA and Tukey HSD for total VOCs emitted, the more detailed analysis with PCA (focused on individual VOCs) accounted for less than 70% variance. This was the case for all but one case (Figure 5) of Frontenac grown in South Dakota in 2013. Statistical analysis using PCA indicated that Frontenac and Marquette were most similar in total VOC emission profile (i.e., clustered around the origin). St. Croix cultivars had a higher positive correlation with the first principal component. Seventeen VOCs with correlation ≥0.300 are listed in a section on statistical analysis. Any differences in the soil and microclimate of these two sites affected overall VOCs emitted from La Crescent and St. Croix cultivars during this research. It is cautioned that these differences could also be affected by the genetics of the cold-hardy hybrids. Frontenac and Marquette share similar parentage [41]. 

The advantages of the modified glass vials over PVF film chambers are its compact design for field sampling, reusability, reduced background contamination from glass, and isolation of VOCs emitted from a single berry. Vacuum-assisted headspace SPME sampling has been used in carefully controlled laboratory settings, to successfully achieve shorter sampling times at lower sampling temperature with good sensitivity and precision to extract polychlorinated biphenyls (PCB) from water [42]. This novel sampling device was the logical next step to isolate VOCs emitted from grape berries during development. This sampling technique is comparable to a viticulturalist ‘smelling’ a grape, and detecting only the volatile compounds emitted through the grape skin. These VOCs are recognized as “free” aroma compounds not bound to a sugar moiety within the berry [43]. This could allow for monitoring of VOCs to measure berry ripeness by instrumental methods. Grapes get softer as they develop, and some cultivars are prone to slip skin (i.e., the grape skin slips easily from the fruit pulp). A disadvantage in using modified glass vials for grape sampling in this research is that increased vacuum was needed as the grapes developed and softened, and sometimes broke the grape skin, more often in the St. Croix cultivar. Another confounding element could be the presence and interference of volatile compounds on the grape skin but not produced by the grape (i.e., pesticide residues, naturally occurring yeasts and molds).

Non-destructive, sampling of VOCs emitted in-vivo from cold-hardy grapes was conducted using 2 methods. PVF film sampling chambers with custom SPME sampling port was used to monitor whole cluster VOC emissions. Modified glass vials supported SPME sampling of individual berries. For comparison to both non-destructive methods, a random 5 berry sample was collected, crushed, and analyzed under controlled laboratory conditions. Statistical analysis using PCA indicated that sampling by PVF chambers and modified glass vials detected similar VOC emission profiles across all 4 cultivars. There was 1 outlier from the glass vial method, indicating a higher than average concentration of styrene in La Crescent grapes. This data could provide evidence of styrene as a product of 2-phenylethanol synthesis from yeast cells [44] (p. 309) present during sampling. It should be noted that 2-phenylethanol variable is positively correlated with principal component 2, orthogonal to styrene. It is expected to have more VOCs detected at higher relative concentrations in crushed berry analysis because of the release of juices and volatiles bound within the berry skin and pulp. Research is warranted to compare headspace SPME analyses of crushed berries with conventional analytical methods such as liquid-liquid extraction [45]. 

Berry VOCs were sampled within a 3-month growing period each year for 2 years. VOC profile in 2012, sampled by PVF chambers and crushed berries were similar in profile. Data points from the PCA fall close to the origin throughout the year in 2012, not shown. This indicated that VOCs collected via PVF chambers did not show noticeable changes from veraison to harvest. VOC profile in 2013, sampled by modified glass vials and crushed berries were similar in profile (points near the origin) until the end of August, with the exception of the outlier on 14 August 2013, Figure 3. In La Crescent and St. Croix cultivars grown in Iowa, there is a movement towards higher than average VOC emission on 24 August 2016, Figure 4. In Iowa, VOC development was still trending above average at harvest on 3 August 2016 for St. Croix and on 29 September 2016 for La Crescent, not shown. VOCs emitted from Frontenac, Marquette, and St Croix cultivars grown in South Dakota started to develop and deviate from average later than Iowa on 29 August 2016, Figure 5. VOCs emitted from La Crescent grown in South Dakota start to trend above average on 3 September 2016, Figure 6. VOC emissions returned to average levels between 29 August and 5 September (harvest) in South Dakota Frontenac berries, Figure 7. The same decreasing trend was observed in South Dakota Marquette between 5 September and 8 September 2016, not shown. Similar to Iowa, the increased VOCs emitted from South Dakota St. Croix and La Crescent do not decline by harvest, not shown.

Differences in microclimate of Iowa and South Dakota plots did not affect VOC emissions from 4 cold-hardy grape cultivars. Little difference in VOC emissions is expected from Marquette and Frontenac because of a shared pedigree. Greater changes in VOC emissions was observed between destructive crushed berry analysis and non-destructive in-vivo analysis methods, but not within the non-destructive methods. In Iowa and South Dakota plots, VOCs emitted from St. Croix and La Crescent cultivars continued to change from veraison through harvest. VOCs emitted in-vivo from Frontenac and Marquette cultivars in South Dakota started to decline 8 days and 3 days before harvest, respectively. More research is warranted in order to make recommendations to viticulturists regarding ideal harvest time for maximum aromas in the cold-hardy grapes. Linking correlations between viticultural practices can enhance the quality of wines for new cold-climate cultivars.

Several improvements to the proposed in-vivo sampling are warranted. Addition of internal standard (IS) [46], for example a small vial with a membrane for controlled emission of IS during sampling (e.g., inside a PVF bag) would to ensure that sampling temperature and SPME fiber variables are controlled. This information would help to normalize sampling variables in field conditions and potentially help with data quality. Secondly, IS addition would enable quantification of volatiles. 

## 4. Materials and Methods 

### 4.1. Overview

A detailed description of Materials and Methods is provided elsewhere [36]. Briefly, below are the summaries of particular approaches used. Research vineyards were located at South Dakota State University (SDSU, Brookings, SD, USA) and Iowa State University (ISU, Ames, IA, USA). Grape clusters were randomly selected, and volatiles from the same clusters were sampled from veraison to harvest. Veraison is defined as when half of the clusters have changed to their ripe color and is shown as the first time point in Results. Collection of volatiles from whole clusters and single berries was completed in 2012 and 2013 seasons, respectively. Berry chemistry data (i.e., Brix, pH, ambient temperatures, and titratable acidity (IA only)) is provided in Appendix A. Volatiles from crushed berries were collected at the same time as in-vivo sampling for both growing seasons. A SPME (65 µm polydimethylsiloxane (PDMS)/divinylbenzene (DVB)) fibers were used for on-site sampling at vineyards and for headspace extraction from crushed berries. No internal standard was used. However, trip blanks (i.e., ambient air samples collected at each vineyard) and sampling vial blanks (for destructive sampling) were used to account for potential interfering volatiles. Four replicates (vines) were sampled per site and cultivar at each time point. 

### 4.2. In-Vivo Sampling of Volatiles from a Whole Cluster of Grapes

Sampling chambers (~5 L volume) for the non-destructive collection of in-vivo volatiles were made from a PVF film and held firm with clean aluminum wire cage framing. Preconditioned (cleaned) PVF chambers were fitted with custom sampling ports for insertion of SPME needles. Typical sampling time was 30 min. 

### 4.3. In-Vivo Sampling of Volatiles from a Single Grape

A standard 2 mL glass vials were modified by removing flat bottoms (Fisher Scientific, Waltham, MA, USA) at a glass shop. The edges of were flared and rounded. A half hole septa was added to the screw top to support the SPME needle. The SPME fiber was placed through the septa prior to sampling. After assembly and placement of the vial apparatus on the individual berry (Appendix A), 5 mL of air was pulled from the vial using a syringe. Care was taken not to disturb the SPME fiber with the syringe needle. The resultant vacuum held the apparatus in place (i.e., sealed by suction into berry surface) while the SPME fiber was exposed for vacuum-assisted VOC sampling. The single berry sampling vials were cleaned prior to each sampling by rinsing in deionized water and oven baked overnight at 107 °C. Cleaned vials were transported in an aluminum lined box. PTFE screw tops were replaced after each sampling.

### 4.4. Destructive Sampling

Berries were collected from each cultivar on the same day and time of in-vivo sampling. Five berries were collected from clusters adjacent to the cluster tagged for in-vivo sampling (i.e., from the same vine but a different cluster than in-vivo sampled berries). Collected berries were frozen prior to analysis and stored in a −20 °C freezer. Berries collected in South Dakota were also frozen and shipped on ice overnight for analysis in Iowa. Frozen berries were hand-crushed in the lab, placed into 20 mL amber screw top vials (Wheaton, Millville, NJ) with PTFE/silicone septa. A CTC CombiPal (LEAP Technologies, Carrboro, NC, USA) was used for automated SPME sampling. Briefly, the vials were agitated and heated to 50 °C for 10 min, followed by 30 min agitated headspace sampling using 65 μm PDMS/DVB SPME fiber. The fiber was thermally desorbed under a flow of helium prior to each sample exposure. These sampling parameters were determined, not shown.

### 4.5. Data Acquisition and Analysis

A custom multidimensional GC was used (Microanalytics, a part of Volatile Analysis Corporation, Round Rock, TX, USA), built on a standard Agilent 6890 platform (Agilent Technologies, Santa Clara, CA, USA). System automation and data acquisition software were MultiTrax (Microanalytics, Round Rock, TX, USA) and ChemStation (Agilent Technologies, Santa Clara, CA, USA). Chromatography was performed on two capillary columns connected in series. The first column was 5% phenyl polysilphenylene-siloxane (30 m × 0.53 mm inner diameter × 0.5 μm thickness, Trajan Scientific, Austin, TX, USA) with a fixed restrictor pre-column. The second polar column was bonded polyethylene glycol in a Sol-Gel matrix (30 m × 0.53 mm inner diameter × 0.5 μm thickness, Trajan Scientific, Austin, TX, USA). The midpoint between the two columns was maintained at a constant pressure of 0.39 atm by a pneumatic switch. In this research, all effluent from the first column was directed into the 2nd analytical column, i.e., no heartcutting was performed. The instrument was also equipped with a flame ionization detector (FID). Flow to the FID can also directed at the midpoint, but FID was not utilized in this research. True multidimensional analyses were not performed, i.e., the system was used in full heartcut mode, meaning separation was performed on both columns in series. Effluent from the second polar column was simultaneously directed to a single quadrupole MS (Model 5973N, Agilent Technologies, Santa Clara, CA, USA) and an olfactometry (sniff) port (Microanalytics, Round Rock, TX, USA) via an open spit interface at atmospheric pressure. The sniff port is equipped with a purge flow controller and supplied with humidified air at 0.54 atm. Flow to the MS and sniff port is determined by fixed restrictor columns, 1 part to MS and 3 to sniff port. Olfactometry was not utilized in the research. The GC inlet was operated in splitless mode at 250 °C. GC oven parameters start with an initial temperature of 40 °C, held for 3.0 min, followed by a 7 °C per min ramp to 240 °C, held for 8.43 min. Total run time was 40 min. Carrier gas is ultra-high purity (UHP) helium (99.999%, Airgas, Des Moines, IA, USA). Temperature of the sniff port and MS transfer lines were 240 °C and 280 °C, respectively. MS full scan range was set from 34 *m*/*z* to 350 *m*/*z*. Scans were collected in electron ionization (EI) mode with an ionization energy of 70 eV. MS heated zones for quadrupole and source were 150 °C and 230 °C, respectively. Daily tuning of the MS was performed with perfluorotributylamine (PFTBA) before each analysis.

Identification of compounds was performed using Automated Mass Spectral Deconvolution and Identification System (AMDIS) target library search with at least 80% mass spectral match. Target libraries included (a) the 6 libraries that are included with the AMDIS program, (b) an onsite library created from analysis of pure standards (200+ compounds), (c) NIST11 mass spectral library. Analysis of variance (ANOVA) was performed using XLSTAT 2016.04.33113 (Addinsoft, New York, NY, USA). The effects of cultivar, site, sampling time, and sampling methods and their interactions on volatiles emitted were analyzed using ANOVA (with confidence interval of 95% and the tolerance of 0.0001) followed by post-hoc (Tukey honestly significant difference, HSD) test. Multivariate analysis was performed using JMP Pro 12.0.1 (SAS Institute Inc., Cary, NC, USA).

## 5. Conclusions

We have shown that is feasible to detect VOCs emitted in-vivo from single grape berries (39 compounds) and whole clusters (44 compounds). Over 110 VOCs were released to headspace from crushed berries. Spatial (vineyard location) and temporal variations in VOC profiles were observed for all four cultivars. However, these changes were not consistent by growing season, by location, within cultivars, by ripening stage when analyzed by multivariate analyses such principal component analysis (PCA) and hierarchical cluster analyses (HCA). Research into aroma compounds present in cold-hardy cultivars is essential to the continued growth of the wine industry in cold climates and diversification of agriculture in the upper Midwestern area of the U.S.

## Figures and Tables

**Figure 1 molecules-24-00536-f001:**
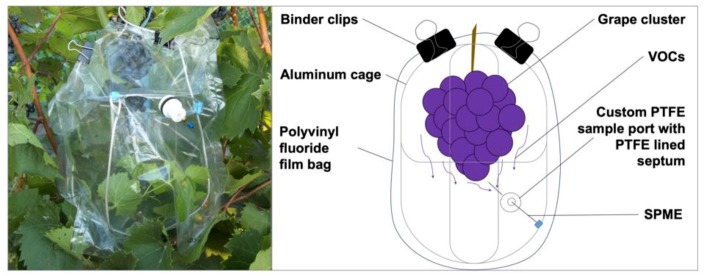
Non-destructive sampling of biogenic volatiles emitted by the whole cluster of grapes on a vine. Schematic of polyvinyl fluoride (PVF) film chambers used for short-term enclosing of growing clusters of cold-hardy grapes during in-vivo sampling of volatile emissions using solid-phase microextraction (SPME). An aluminum wire cage was constructed to hold the PVF chamber spread around and to be secured to the grape vine’s training system. The PVF chamber was modified with a custom polytetrafluoroethylene (PTFE) port fitted with 11 mm PTFE lined silicone septa (SPME sampling port).

**Figure 2 molecules-24-00536-f002:**
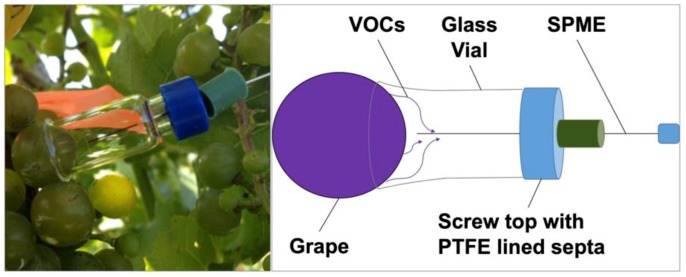
Non-destructive sampling of biogenic volatiles emitted by a modified 2 mL glass vial for a vacuum-assisted sampling of volatiles from a single grape berry. Schematic of a modified screw top 2 mL glass vial with PTFE lined septa used for characterizing in-vivo metabolite emissions from selected cold-hardy grapes. Negative pressure was created with a syringe to hold the sampling device with SPME sealed onto the grape berry surface.

**Figure 3 molecules-24-00536-f003:**
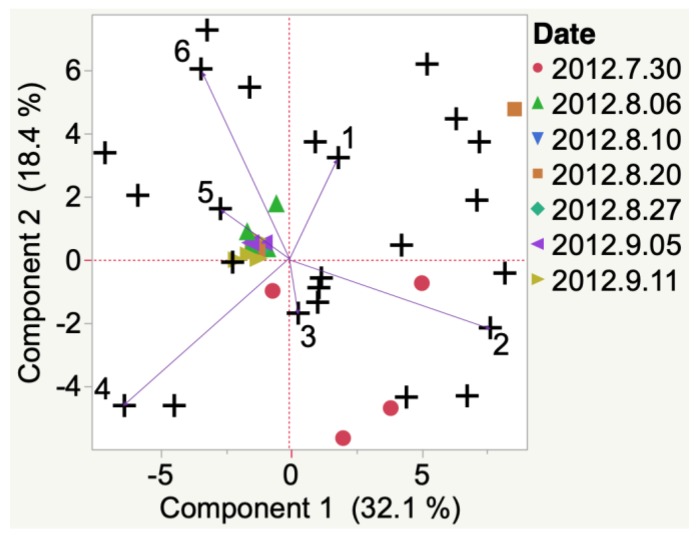
Evolution of VOCs emitted from whole clusters of 2012 Frontenac grapes grown in South Dakota from veraison to harvest. Key to most representative volatiles from HCA, shown as vectors from the origin and read clockwise: 1 = 2-Methyl-3-penten-2-one, 2 = Nonanal, 3 = Benzyl alcohol, 4 = Toluene, 5 = Acetic acid, 6 = 3-Phenyl-2-propenal.

**Figure 4 molecules-24-00536-f004:**
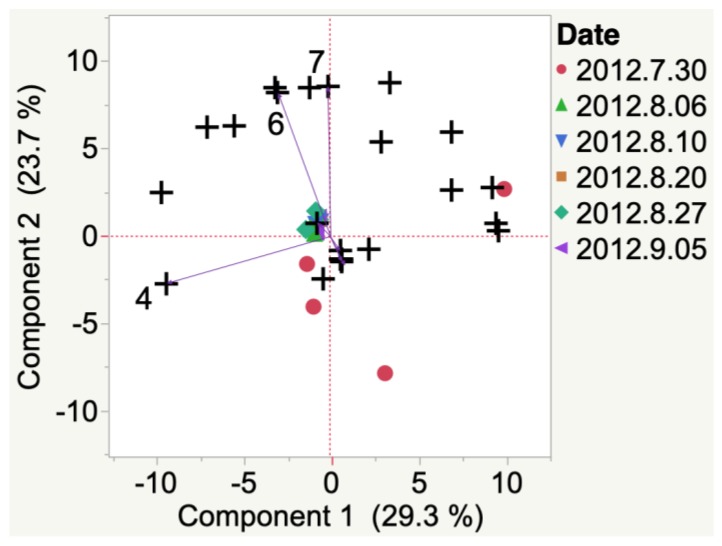
Evolution of VOCs emitted from whole clusters of 2012 Marquette grapes grown in South Dakota from veraison to harvest. Key to most representative volatiles from HCA, shown as vectors from the origin and read clockwise: 1 = 1-Hexadecanol, 2 = 2-Ethyl-1-hexanol, 3 = 1-Pentanol, 4 = Acetone, 5 = Indene, 6 = Decane, 7 = 4-Methyl-3-penten-2-one.

**Figure 5 molecules-24-00536-f005:**
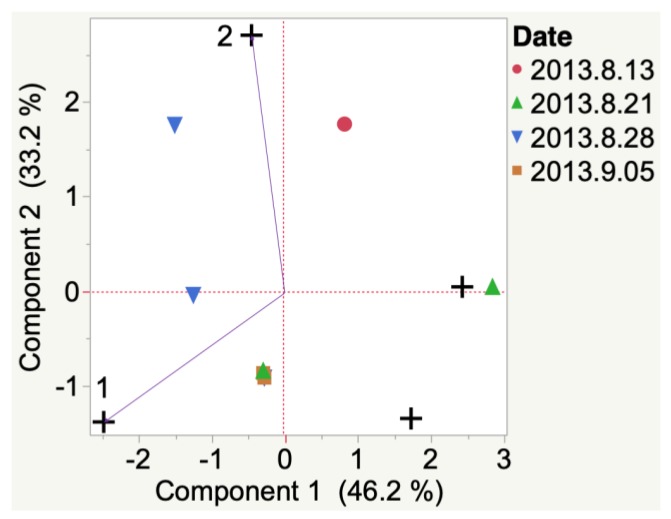
Evolution of VOCs emitted from single berry from veraison to harvest of 2013 Frontenac grapes grown in South Dakota. Key to most representative volatiles from HCA, shown as vectors from the origin and read clockwise: 1 = Palmitic acid, 2 = Acetic acid.

**Figure 6 molecules-24-00536-f006:**
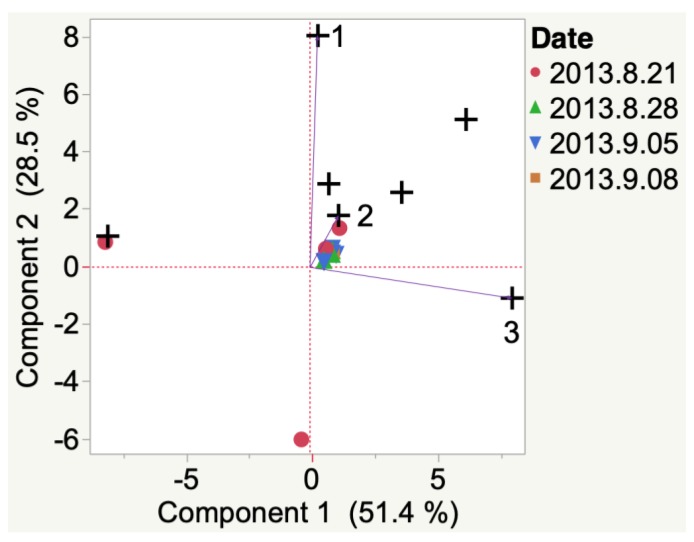
Evolution of VOCs emitted from single berry from veraison to harvest of 2013 Marquette grapes grown in South Dakota. Key to most representative volatiles from HCA, shown as vectors from the origin and read clockwise: 1 = Ethyl octanoate, 2 = 2-Ethyl-1-hexanol, 3 = 1,4-Butanolide.

**Figure 7 molecules-24-00536-f007:**
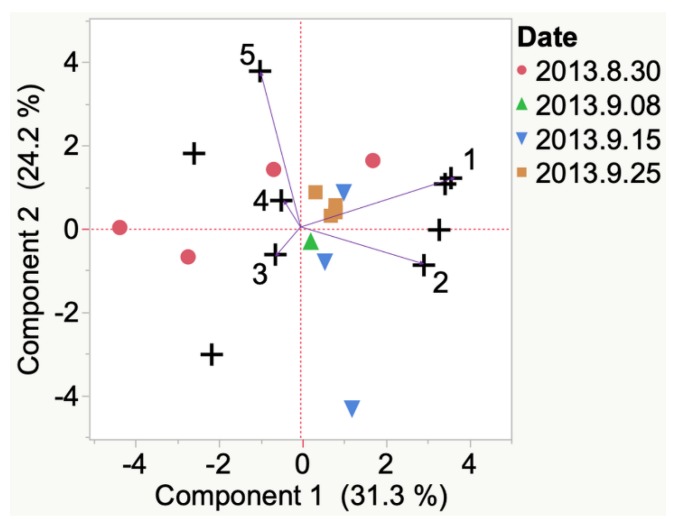
Evolution of VOCs emitted from single berry from veraison to harvest of 2013 St. Croix grapes grown in South Dakota. Key to most representative volatiles from HCA, shown as vectors from the origin and read clockwise: 1 = Nonanal, 2 = Diacetone alcohol, 3 = 1,4-Butanolide, 4 = 5-(Hydroxymethyl)-1-furancarboxaldehyde, 5 = Ethyl acetate.

**Figure 8 molecules-24-00536-f008:**
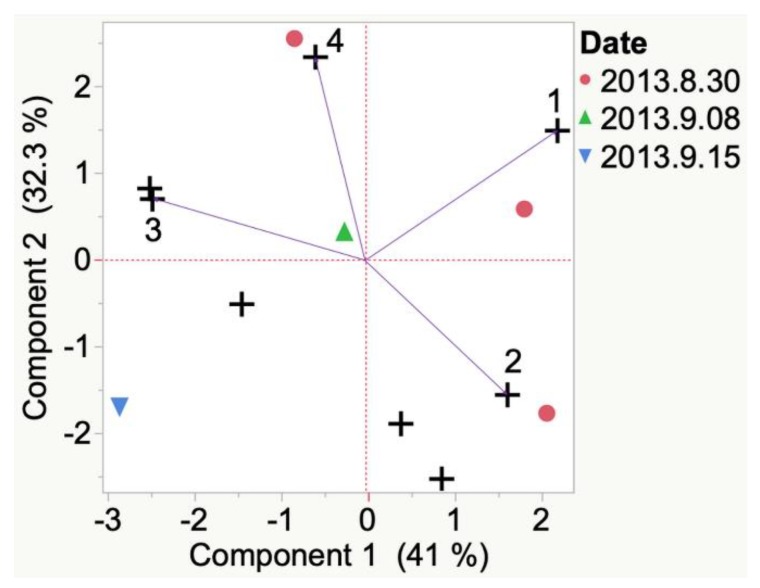
Evolution of VOCs emitted from single berry from veraison to harvest of 2013 St. Croix grapes grown in South Dakota. Key to most representative volatiles from HCA, shown as vectors from the origin and read clockwise: 1 = 2-Phenylethanol, 3 = Diacetone alcohol, 4 = Acetic acid, 2 = 6-Methyl-5-hepten-2-one.

**Figure 9 molecules-24-00536-f009:**
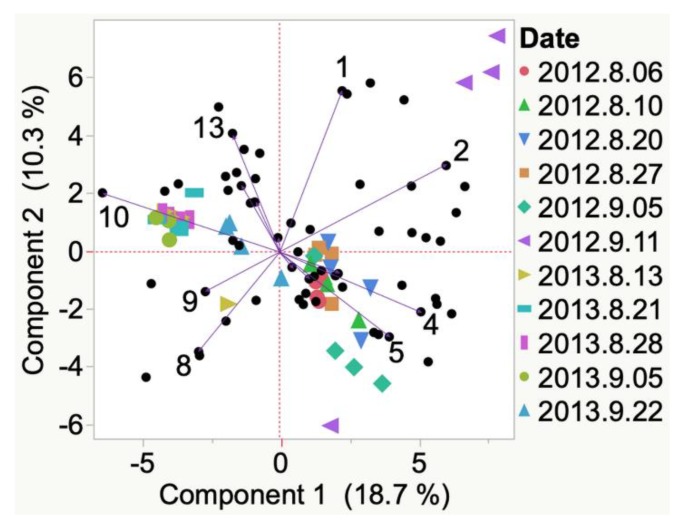
Evolution of VOCs emitted from crushed berries from veraison to harvest of 2012 and 2013 Frontenac grapes grown in South Dakota. Key to most representative volatiles from HCA, shown as vectors from the origin and read clockwise: 1 = Nonane, 2 = Styrene, 3 = Octanal, 4 = Acetaldehyde, 5 = 1-Hexanol, 6 = Benzoic acid, methyl ester, 7 = Isophorone, 8 = Ethyl hexanoate, 9 = *N*-benzyl-2-phenethylamine, 10 = Acetone, 11 = Ethyl palmitate, 12 = Hexanoic acid, 13 = 2-Methyl-1-propanol, 14 = 2-Octanone.

**Figure 10 molecules-24-00536-f010:**
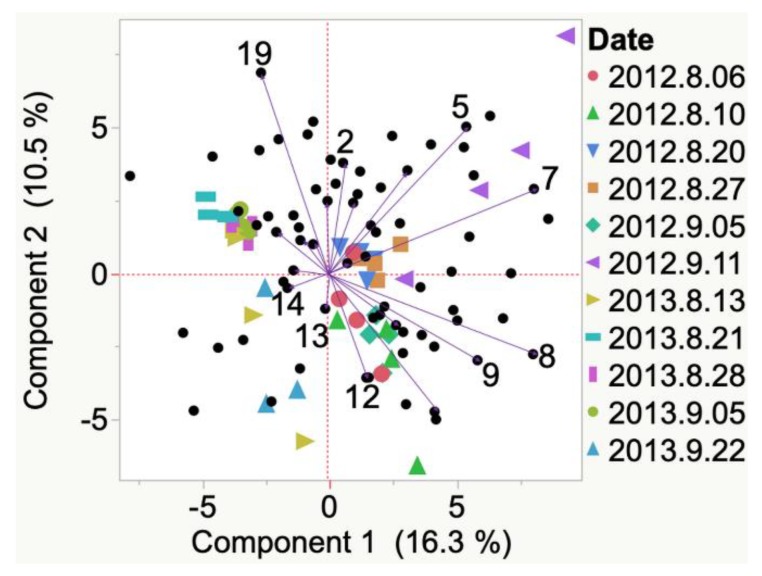
Evolution of VOCs emitted from crushed berries from veraison to harvest of 2012 and 2013 Marquette grapes grown in South Dakota. Key to most representative volatiles from HCA, shown as vectors from the origin and read clockwise: 1 = 1-Heptanol, 2 = Amyl acetate, 3 = Methyl ethyl ketone, 4 = Decane, 5 = Styrene, 6 = beta-Cyclocitral, 7 = Nonanal, 8 = Acetaldehyde, 9 = Valeraldehyde, 10 = Octanal, 11 = Cyclohexanol, 12 = (*E*)-2-hexenoic acid, 13 = Methyl disulfide, 14 = Nonane, 15 = Allyl alcohol, 16 = beta-Damascenone, 17 = Nerol acetate, 18 = *p*-cymene, 19 = 1-Pentanol.

**Figure 11 molecules-24-00536-f011:**
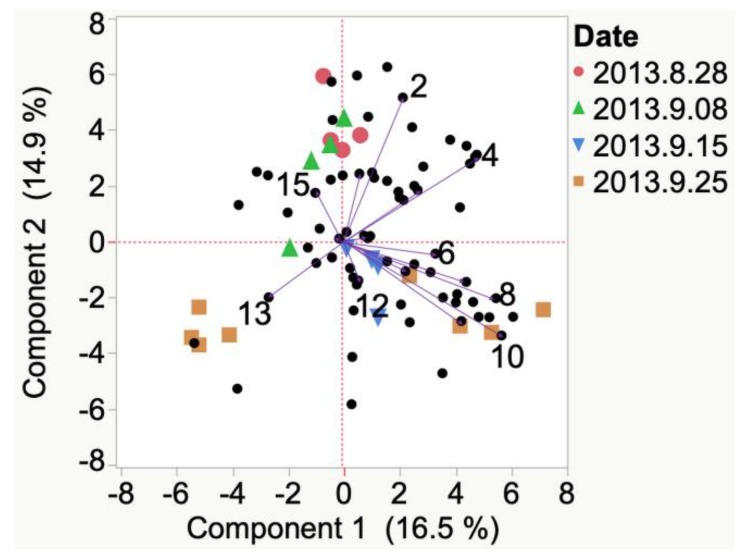
Evolution of VOCs emitted from crushed berries from veraison to harvest of 2013 St. Croix grapes grown in South Dakota. Key to most representative volatiles from HCA, shown as vectors from the origin and read clockwise: 1 = Benzyl alcohol, 2 = Benzaldehyde, 3 = Octanal, 4 = Acetophenone, 5 = Linalool, 6 = Ethyl acetate, 7 = Methyl salicylate, 8 = Safrol, 9 = Propionaldehyde, 10 = 2-Phenylethanol, 11 = 1-Pentanol, 12 = 2-Heptanone, 13 = Benzoic acid, methyl ester, 14 = Aspirin methyl ester, 15 = *N*-benzyl-2-phenethylamine.

**Figure 12 molecules-24-00536-f012:**
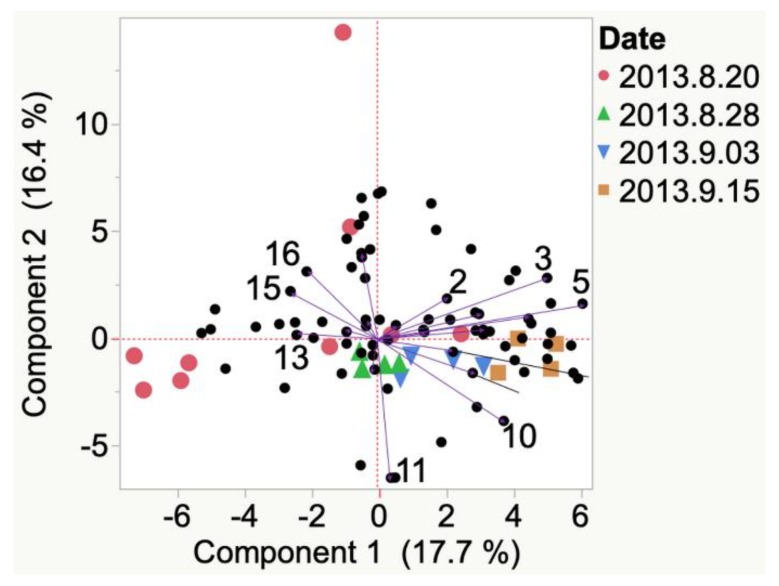
Evolution of VOCs emitted from crushed berries from veraison to harvest of 2012 and 2013 La Crescent grapes grown in South Dakota. Key to most representative volatiles from HCA, shown as vectors from the origin and read clockwise: 1 = Furfural, 2 = Isobutyraldehyde, 3 = Benzaldehyde, 4 = Ethyl vinyl ketone, 5 = Toluene, 6 = Camphene, 7 = Ethyl butyrate, 8 = Propyl-Benzene, 9 = Styrene, 10 = Hexanal, 11 = beta-Pinene, 12 = Isoamyl acetate, 13 = Carbon disulfide, 14 = Linalyl acetate, 15 = 2-Ethyl-1-hexanol, 16 = Geraniol, 17 = Isophorone, 18 = Allyl alcohol.

**Table 1 molecules-24-00536-t001:** Whole cluster analysis. Volatiles emitted from Frontenac and Marquette clusters, grown in Iowa (IA) and South Dakota (SD). These VOCs were indicated to be the most representative variable from hierarchical clustering analysis (HCA) after the PCA (JMP Pro 12.0.1, SAS Institute Inc., Cary, NC, USA).

Sample	Cluster	No. of Members ^2^	Most Representative Variable ^3^	Cluster Proportion of Variation Explained ^4^	Total Proportion of Variation Explained ^5^
IA Frontenac (0.709) ^1^	1	5	Heptanal	0.937	0.173
	3	5	4-Methyl-3-penten-2-one	0.745	0.138
	5	6	Nonanal	0.512	0.114
	2	3	3-Methyl-1-butanol	0.845	0.094
	4	4	1,4-Butanolide	0.576	0.085
	6	2	5-(Hydroxymethyl)-2-furancarboxaldehyde	0.858	0.064
	7	2	Benzophenone	0.548	0.041
SD Frontenac (0.686) ^1^	1	7	Toluene	0.712	0.208
	2	5	Nonanal	0.820	0.171
	5	3	3-Phenyl-2-propenal	0.810	0.101
	3	4	4-Methyl-3-penten-2-one	0.592	0.099
	4	3	Acetic acid	0.502	0.063
	6	2	Benzyl alcohol	0.536	0.045
IA Marquette (0.739) ^1^	1	10	1-Octanol	0.805	0.310
	2	5	Acetaldehyde	0.814	0.156
	3	3	Methyl ethyl ketone	0.616	0.071
	4	3	1-Hexadecanol	0.505	0.063
	5	3	Acetophenone	0.536	0.062
	6	1	Acetic acid	1.000	0.038
	7	1	2-Ethyl-1-hexanol	1.000	0.038
SD Marquette (0.783) ^1^	1	7	Acetone	0.817	0.249
	2	4	4-Methyl-3-penten-2-one	0.799	0.139
	6	4	Decane	0.797	0.139
	3	2	1-Pentanol	1.000	0.087
	4	3	2-Ethyl-1-hexanol	0.532	0.069
	7	2	1-Hexadecanol	0.654	0.057
	5	1	Indene	1.000	0.043

^1^ The total proportion of variation explained by all the cluster components. ^2^ The number of variables in the cluster. ^3^ The cluster variable that has the largest squared correlation with its cluster component. ^4^ The cluster’s proportion of variance explained by the first principal component amount the variables in the cluster, based only on variables within the cluster. ^5^ The overall proportion of variance explained by the cluster component, using only the variables within each cluster to calculate the first principal component.

**Table 2 molecules-24-00536-t002:** ‘Characteristic’ VOCs emitted from single berries of Frontenac, Marquette, St. Croix, and La Crescent grapes grown in Iowa and South Dakota. These VOCs were indicated to be the most representative variable from hierarchical clustering analysis after the PCA (JMP Pro 12.0.1, SAS Institute Inc., Cary, NC, USA).

Sample	Cluster	No. of Members ^2^	Most Representative Variable ^3^	Cluster Proportion of Variation Explained ^4^	Total Proportion of Variation Explained ^5^
SD Frontenac (0.750) ^1^	1	2	Palmitic acid	0.830	0.415
	2	2	Acetic acid	0.669	0.334
SD Marquette (0.870) ^1^	1	7	1,4-Butanolide	0.878	0.473
	2	4	Ethyl octanoate	0.883	0.272
	3	2	2-Ethyl-1-hexanol	0.819	0.126
IA St. Croix (0.896) ^1^	1	4	3-Methyl indole	1.000	0.500
	2	3	Acetic acid	0.722	0.271
	3	1	Benzyl alcohol	1.000	0.125
SD St. Croix (0.855) ^1^	1	3	Nonanal	0.802	0.241
	3	3	Diacetone alcohol	0.713	0.214
	2	2	1,4-Butanolide	1.000	0.200
	4	1	5-(Hydroxymethyl)-2-furancarboxaldehyde	1.000	0.100
	5	1	Ethyl acetate	1.000	0.100
IA La Crescent (0.909) ^1^	1	28	2-Ethyl-1-hexanol	0.973	0.757
	2	4	3-Methyl indole	0.772	0.086
	4	2	Ethanol	0.636	0.035
	3	2	2-Phenylethanol	0.556	0.031
SD La Crescent (0.936) ^1^	1	3	2-Phenylethanol	1.000	0.300
	3	3	Diacetone alcohol	0.853	0.256
	4	2	Acetic acid	0.976	0.195
	2	2	6-Methyl-5-hepten-2-one	0.926	0.185

^1^ The total proportion of variation explained by all the cluster components. ^2^ The number of variables in the cluster. ^3^ The cluster variable that has the largest squared correlation with its cluster component. ^4^ The cluster’s proportion of variance explained by the first principal component amount the variables in the cluster, based only on variables within the cluster. ^5^ The overall proportion of variance explained by the cluster component, using only the variables within each cluster to calculate the first principal component.

**Table 3 molecules-24-00536-t003:** ‘Characteristic’ VOCs emitted from crushed berries of Frontenac, Marquette, St. Croix, and La Crescent grapes grown in Iowa and South Dakota. These VOCs were indicated to be the most representative variable from hierarchical clustering analysis after the PCA (JMP Pro 12.0.1, SAS Institute Inc., Cary, NC, USA).

Sample	Cluster	No. of Members ^B^	Most Representative Variable ^C^	Cluster Proportion of Variation Explained ^D^	Total Proportion of Variation Explained ^E^
IA Frontenac (0.803) ^A^	1	7	3-Methyl-1-butanol	0.993	0.257
	2	9	Cyclohexanol	0.770	0.257
	4	4	Isoamyl acetate	0.745	0.110
	3	4	Isovaleraldehyde	0.622	0.092
	5	3	Toluene	0.774	0.086
SD Frontenac (0.627) ^A^	2	8	Styrene	0.636	0.083
	10	7	Acetaldehyde	0.546	0.063
	5	6	2-Octanone	0.608	0.060
	1	6	Acetone	0.602	0.059
	6	5	1-Hexanol	0.645	0.053
	4	5	Nonane	0.624	0.051
	8	4	Ethyl hexanoate	0.732	0.048
	3	4	Ethyl palmitate	0.694	0.045
	7	3	Hexanoic acid	0.740	0.036
	9	4	*N*-benzyl-2-phenethylamine	0.509	0.033
	12	2	2-Methyl-1-propanol	0.953	0.031
	11	3	Benzoic acid, methyl ester	0.560	0.028
	14	2	Isophorone	0.569	0.019
	13	2	Octanal	0.524	0.017
IA Marquette (0.863) ^A^	1	9	Hexanal	0.925	0.347
	3	5	Isoamyl acetate	0.878	0.183
	2	4	Styrene	0.776	0.129
	5	2	Ethanol	0.933	0.078
	6	2	Benzophenone	0.813	0.068
	4	2	Allyl alcohol	0.703	0.059
SD Marquette (0.654) ^A^	7	7	Acetaldehyde	0.617	0.062
	6	6	Methyl ethyl ketone	0.639	0.055
	3	5	Decane	0.760	0.054
	19	5	Nonanal	0.667	0.048
	1	4	Styrene	0.777	0.044
	5	5	Amyl acetate	0.621	0.044
	4	4	(*E*)-2-Hexenoic acid	0.704	0.040
	10	4	Cyclohexanol	0.692	0.040
	9	5	Octanal	0.480	0.034
	2	3	1-Pentanol	0.673	0.029
	8	2	Nonane	0.966	0.028
	18	3	Valeraldehyde	0.630	0.027
	11	3	1-Heptanol	0.629	0.027
	14	4	beta-Damascenone	0.470	0.027
	12	4	Allyl alcohol	0.435	0.025
	13	2	*p*-Cymene	0.835	0.024
	16	2	Methyl disulfide	0.635	0.018
	15	1	beta-Cyclocitral	1.000	0.014
	17	1	Nerol acetate	1.000	0.014
IA St. Croix (0.772) ^A^	1	9	Formic acid, octyl ester	0.832	0.150
	4	8	Ethyl decanoate	0.901	0.144
	2	7	Isobutyraldehyde	0.674	0.094
	3	5	Aspirin methyl ester	0.813	0.081
	5	3	Benzeneacetaldehyde	0.858	0.052
	10	3	Ethanol	0.771	0.046
	8	3	Methacrolein	0.682	0.041
	12	2	Isoamyl acetate	0.841	0.034
	6	2	1-Butanol	0.790	0.032
	7	3	Ethyl butyrate	0.493	0.030
	9	2	1-Hexanol	0.649	0.026
	11	2	beta-Damascenone	0.576	0.023
	13	1	Valeraldehyde	1.000	0.020
SD St. Croix (0.692) ^A^	2	8	Acetophenone	0.685	0.081
	3	6	Linalool	0.786	0.069
	6	6	Benzaldehyde	0.727	0.064
	7	6	Methyl salicylate	0.681	0.060
	5	6	Cyclohexanol	0.662	0.058
	4	6	2-Heptanone	0.627	0.055
	1	5	2-Phenylethanol	0.728	0.054
	10	5	1-Pentanol	0.581	0.043
	8	5	Benzyl alcohol	0.557	0.041
	11	3	Safrol	0.855	0.038
	9	3	Benzoic acid, methyl ester	0.797	0.035
	12	3	Ethyl acetate	0.622	0.027
	14	2	Aspirin methyl ester	0.840	0.025
	13	2	Propionaldehyde	0.797	0.023
	15	2	*N*-Benzyl-2-phenethylamine	0.628	0.018
IA La Crescent (0.699) ^A^	1	11	beta-Cyclocitral	0.682	0.121
	2	8	beta-Pinene	0.836	0.108
	3	9	Ethyl butyrate	0.710	0.103
	8	4	*p*-Cymene	0.663	0.043
	9	4	Propanoic acid	0.648	0.042
	12	3	1-Hexanol	0.792	0.038
	6	3	Nerol acetate	0.776	0.038
	4	4	Methacrolein	0.565	0.036
	5	4	Beta-damascenone	0.521	0.034
	7	3	(+)-4-Carene	0.650	0.031
	11	3	Valeric acid	0.641	0.031
	13	3	3-Methyl-1-butanol	0.638	0.031
	10	2	Acetic acid	0.845	0.027
	14	1	Propyl-benzene	1.000	0.016
SD La Crescent (0.741) ^A^	3	8	Allyl alcohol	0.845	0.086
	2	8	beta-Pinene	0.837	0.085
	1	7	Toluene	0.691	0.061
	11	5	Isoamyl acetate	0.915	0.058
	6	6	Isophorone	0.637	0.048
	7	6	Ethyl butyrate	0.567	0.043
	8	5	Hexanal	0.669	0.042
	4	5	Benzaldehyde	0.657	0.042
	13	5	Styrene	0.618	0.039
	15	4	Carbon disulfide	0.771	0.039
	9	3	Ethyl vinyl ketone	1.000	0.038
	5	3	Camphene	0.900	0.034
	18	3	Linalyl acetate	0.806	0.031
	17	3	Geraniol	0.730	0.028
	10	2	Furfural	0.908	0.023
	12	3	Isobutyraldehyde	0.499	0.019
	16	2	2-Ethyl-1-hexanol	0.500	0.013
	14	1	Propyl-benzene	1.000	0.013

^A^ The total proportion of variation explained by all the cluster components. ^B^ The number of variables in the cluster. ^C^ The cluster variable that has the largest squared correlation with its cluster component. ^D^ The cluster’s proportion of variance explained by the first principal component amount the variables in the cluster, based only on variables within the cluster. ^E^ The overall proportion of variance explained by the cluster component, using only the variables within each cluster to calculate the first principal component.

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
