# Peer review of "Evaluation of Volatile Metabolites Emitted In-Vivo from Cold-Hardy Grapes during Ripening Using SPME and GC-MS: A Proof-of-Concept"

_molecules, 2019, doi:10.3390/molecules24030536_

Reviewer 1 Report

The authors present interesting approaches for monitoring volatiles during grape ripening. Some revisions are needed to improve the clarity of the manuscript. The experimental design would be improved by ensuring that some type of internal standard were incorporated into the sampling and by correcting for ambient volatiles at each sampling point. See also specific comments below.

Introduction

--Please write out the objectives in full sentence format (lines 124-139).

Results

--It is not appropriate to apply the sensory descriptors for the grapes as done in this manuscript. The descriptors may be appropriate for the pure chemical compounds listed, however that does not mean that the grapes will have this aroma when smelled. Please remove all of these descriptors and discuss only the changes in chemical composition. If the authors actually have sensory descriptive analysis of the grapes at the different ripening stages, that could provide interesting information but would require further statistical analyses (e.g., PLS) to relate chemical composition and sensory properties.

--For many of the grapes, no real differences are observed in the volatile profile with ripening according to the figures (eg.g., Figures 3, 4, 6). This is not mentioned until much later in the discussion, but should be first noted in the Results section. Possibly the first time point should be removed from the analysis and then the analysis (PCA?) redone to see if changes are observed. ANOVA followed by a careful review of trends in volatile profiles that are statistically significant across ripening stages may also reveal changes that occur during ripening.

--Figures 9, 10, 11, 12 are extremely difficult to interpret. Could the colors for the samples at the different time point be coded so that visually they could go from something more associated with 'unripe' grapes (e.g., green) to 'ripe' grapes (e.g.,blue/purple) (I realize that white grapes like La Crescent will not be purple at harvest). Further, in some cases, specific volatiles are noted in the text and in the figure caption, but then they are not identified in the figure (see for example Figure 4). Maybe volatile numbers could be used in the actual figures instead of + signs and black dots.

--Is there any information on physiologic maturity at the different sampling times (e.g., Brix, pH, TA, size, color, EL-stage, etc.? Some information relating the volatile composition to physiologic maturity would be helpful.

Materials and Methods

--the authors refer to veraison in the manuscript but they do not define/identify the time point for veraison; this information should be provided in the materials and methods and also in the figures so that the reader can clearly identify this time point during the ripening period

--How long prior to SPME sampling was the PVF film placed around the grape clusters for equilibration of volatiles into the vapor phase? Did any moisture collect in the bag due to respiration? What was the temperature range during each sampling? (lines 438-442)

--How long was SPME sampling for the single berry sampling? How long prior to SPME sample was the vial placed around the grape for equilibration of volatiles into the vapor phase? Did any moisture collect in the vial due to respiration? What was the temperature range during each sampling? (lines 443-453.

--For all samples, was a background air control sample in the vineyard obtained to ensure that measured volatiles were from the grape and not from from ambient environment. Having this information would help to clarify some of the comments about some compounds coming from exogenous sources. Presumably information on pesticide application is available for each of the vineyards and this information could be used to inform some of the discussion also.

--Incorporation of some type of post-sampling internal standard as per published work (Risticevic et al., Nature Protocols, 5, 162-176, 2010) would help to at least provide some information on relative concentrations of the volatiles. Incorporation of a small vial of some type of internal standard during in vivo sampling is also needed to ensure that sampling temperature and fiber variables are controlled. Without this information it is difficult to know if the reported results are due to sampling variables or to actual changes during maturation.

--For all analyses, how many replications were performed? 

--More information on statistical analyses are needed (line 498). Was an ANOVA done on the volatiles to identify statistically significant volatiles at the different time points, prior to cluster analysis and PCA? Was PCA performed--are PCA plots what is shown in the figures? Was there any data transformation for the multivariate analyses? How were missing values reported in the multivariate analyses?

Author Response

Thank you for the opportunity to address all comments (attached in Word file).

Reviewer 2 Report

The objective of this study was to perform the Evaluation of volatile metabolites emitted in-vivo from cold-hardy grapes during ripening using SPME and GC-MS. The work is interesting and presents merit for publication, however, it needs a better description in the methodology and discuss the results obtained.

Specific comments:

Lines 90-94: Also comment on emerging sample preparation techniques such as LLE for grape and wine analysis. For example: https://doi.org/10.1016/j.foodres.2018.03.020

Lines 124-139: The objectives should be briefly described in a paragraph, not subdivided into several items like the one presented.

Item 4.5: it is not clear whether the method was developed by the authors or used a methodology described in the literature. Basic validation parameters need to be mentioned: R2, LOD, LOQ and recovery% ranges.

Line 496: It was mentioned that a library constructed with more than 200 external standards, but the origin of the standards and the compounds that were analyzed were not mentioned.

Figures 3-8: ACP is a multivariate analysis that requires a large amount of data points. The presented figures were constructed with little data points for an ACP. We also highlight the small percentage of variance explained in PCA.

In general, the work has many results, but the discussion is poor.

Author Response

Comments and Suggestions for Authors

The objective of this study was to perform the Evaluation of volatile metabolites emitted in-vivo from cold-hardy grapes during ripening using SPME and GC-MS. The work is interesting and presents merit for publication, however, it needs a better description in the methodology and discuss the results obtained.

Authors Response: We improved the Methods section and discussion of Results by addressing Reviewer 1. We also provided detailed description of Methods in data paper [ref. 36] now accepted (January 24, 2019). 

Specific comments:

Lines 90-94: Also comment on emerging sample preparation techniques such as LLE for grape and wine analysis. For example: https://doi.org/10.1016/j.foodres.2018.03.020

Authors Response: We added the proposed example of a recent use of conventional approach to analysis of volatiles as reference [45] and added it in the Discussion section when commenting on possible improvements of crushed berries analysis:

“Research is warranted to compare headspace SPME analyses of crashed berries with conventional analytical methods such as liquid-liquid extraction [45].”

Lines 124-139: The objectives should be briefly described in a paragraph, not subdivided into several items like the one presented.

Authors Response: We corrected this and condensed the objectives statement to lines 124-132. Reviewer 1 had a similar suggestion.

Item 4.5: it is not clear whether the method was developed by the authors or used a methodology described in the literature. Basic validation parameters need to be mentioned: R2, LOD, LOQ and recovery% ranges.

Authors Response: Method development is covered in the data paper [36] (now accepted, January 24, 2019). However, full quantification was not developed at this time.  We added quantification of volatiles as warranted in future research in the last paragraph of Discussion.

Line 496: It was mentioned that a library constructed with more than 200 external standards, but the origin of the standards and the compounds that were analyzed were not mentioned.

Authors Response: Listing of all 200+ chemical standards is not feasible.  However, the overwhelming majority of all chemical standards were ACS grade (or similar) and purchased from Sigma-Aldrich, Fluka, or Fisher.

Figures 3-8: ACP is a multivariate analysis that requires a large amount of data points. The presented figures were constructed with little data points for an ACP. We also highlight the small percentage of variance explained in PCA.

Authors Response: Excellent point. It was also brought up by Reviewer 1. We added the first overall mention and discussion at the end of the Section 2.1 (Results) addressing the comment about no real differences for all but one case presented: ‘It should be noted that significant changes in volatiles emitted were only observed in Frontenac grapes grown in South Dakota in 2013 as indicated by the variance accounted for in component 1 and 2 in PCA (i.e., greater than 70%). However, due to the exploratory nature of this research, all PCA data is presented is subsequent sections.’

Regarding ANOVA analyses -  we added ANOVA and post-hoc Tukey HSD analyses (Tables A1, A2 in Appendix A & Table S1 in Supplementary Material).  We added description of ANOVA and post-hoc Tukey HSD in Methods (section 4.5).  In addition, the following summary was added to Discussion:

‘Preliminary analyses using ANOVA were also completed (shown in Table S1, Supplementary Material). Type 1 sum of squares analysis indicated statistical significance of method, cultivar, date & time, method & cultivar interaction, and method & site & cultivar interactions (shown in Table A1). Post-hoc Tukey HSD test is (Table A2) shows differences within method, cultivar, date & time, method & cultivar interaction, and method & site & cultivar interactions. PCA analyses were used to determine key volatile compounds emitted. Although significant differences were noted using ANOVA and Tukey HSD for total VOCs emitted, the more detailed analysis with PCA (focused on individual VOCs) accounted for less than 70% variance. This was the case for all but one case (Figure 5) of Frontenac grown in South Dakota in 2013.’

In general, the work has many results, but the discussion is poor.

Authors Response: we made major revisions to the manuscript, including improvements to the discussion of Results and the Discussion section.

Round  2

Reviewer 1 Report

The authors have revised the manuscript according to review comments.

Reviewer 2 Report

With the changes made the article is suitable for publication.